# Registration by tracking for sequential 2D imaging

## Abstract

In this paper we present a new modality-agnostic image registration method based on sequential 2D imaging. Our method makes use of a tracking framework in combination with a learned sparse-to-dense interpolation scheme. For tracking we employ a Discriminative Correlation Filter (DCF), a fast method that has lately proven extremely useful for visual tracking. Based on displacement vectors from several trackers the dense displacement field is estimated by a neural network using normalized convolutions.

**Keywords:** Image registration, learning methods, visual tracking.

## 1. Introduction

In general, iterative image registration algorithms are computational costly which is problematic in real-time scenarios. In an image registration problem, the motion vectors from one image to the next is referred to as the displacement field. While conventional methods often minimizes some metric based on static points in space (control points), we define the displacement field from a sparse set of tracking points where the spatial location of these points is constantly updated. In the next section, we describe the tracking framework and how a dense displacement field can be estimated from the location of these tracking points.

## 2. Method

Our method contains two modules: a point tracking algorithm and a sparse-to-dense interpolation scheme. The two modules are independent of each other and can therefore easily be replaced or updated separately. The first part: tracking, consists of several trackers where each tracker follows a predefined point, or region, in the image sequence. All translation vectors creates a sparse representation which is used to compute an estimate of the dense displacement field.

### 2.1. Tracking

As trackers, we make use of discriminative correlation filters (DCF) (Danelljan, 2018), which have lately proven extremely useful for visual tracking (Danelljan et al., 2016). The idea is to train correlation filters $f$ that distinguish the tracked target from the background. Each image patch $x$ has a score variable $y$ associated to it. A low score, $y \approx 0$, represents background whereas a high score, $y \approx 1$, represents a target. The filter is initially trained on a set of image patches, $\{x_i\}_{i=1}^N$, wherein the target and the desired output $y_i$ are defined. In practice it is convenient to generate data from the first frame by various augmentation

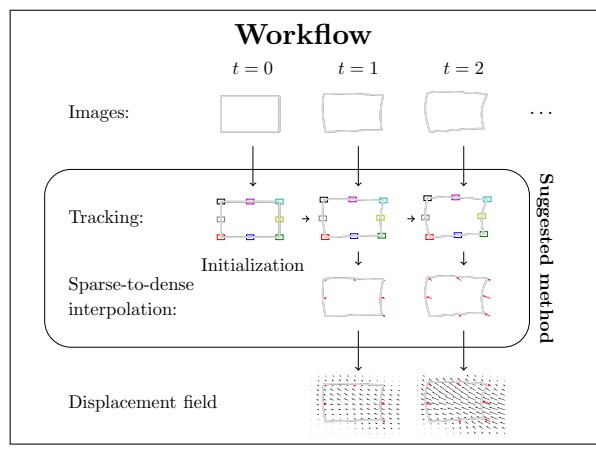
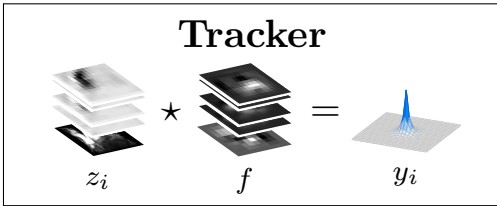
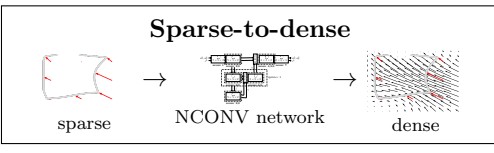

Figure 1: Overview of suggested workflow (left). Our method consists of several trackers where each tracker is a learnt collection of DCFs (upper right). The translation vectors from all trackers are used as a sparse representation of the displacement field and feed into the sparse-to-dense interpolation network (bottom right).

techniques. The filters can either be trained on the image patch itself or on features derived from the image $\{z_i^d\}_{d=1}^D$. The filter coefficients are determined by solving the following regularized linear least-square problem

$$f^\star = \arg\min_f \sum_i^N \alpha_i \left\| \sum_{d=1}^D f^d \star z_i^d - y_i \right\|^2 + \lambda \sum_{d=1}^D ||f^d||^2 \tag{1}$$

where $\alpha_i$ is a hyperparameter controlling the impact of training sample $i$. Recently, the DCF framework was reformulated in the continuous domain to support multi-resolution feature maps without explicit resampling. In this paper we make use of the C-COT tracker (Danelljan et al., 2016) which employs deep convolutional feature maps from already pre-trained networks.

Once trained, the filters, $\{f_r\}_{d=1}^D$ are used to predict scores by convolving it with feature maps, $\{z_i^d\}_{d=1}^D$ from subsequent images. To reduce the computational cost, the operation is performed in the Fourier domain using the Fast Fourier Transform (FFT) and the convolution theorem,

$$\hat{y}_r = \sum_{d=1}^D z^d \star f_r^d = \mathcal{F}^{-1} \left\{ \sum_{d=1}^D \mathcal{F}\{z^d\} \odot \mathcal{F}\{f_r^d\} \right\} \tag{2}$$

For each tracker, a displacement vector is extracted from the target location corresponding to the highest score.

### 2.2. Interpolation

For interpolation, we make use of a sparse-to-dense scheme based on normalized convolutions (Knutsson and Westin, 1993). Traditional convolutions are well-suited for regularly sampled

data, but struggle with irregularly sampled data. Normalized convolution uses confidence map to explain the spatial locations and its certainty. A dense signal can be estimated from a sparse representation based on an applicability function and some basis functions.

The interpolation scheme is designed as an image-to-image neural network with normalized convolutional layers. The network is trained offline by optimizing a loss function that simultaneously minimizes the estimation error and maximizes the output confidence (Eldesokey et al., 2019). Due to the lack of real data we trained using synthetic displacement fields, i.e. without any image data. With diffeomorphic assumptions, we synthesized displacement fields using geodesic shooting (Miller et al., 2006) and emulated sparse representations and confidence maps by sampling coordinates from the generated data. To avoid overfitting new samples are generated at every iteration.

## 3. Results

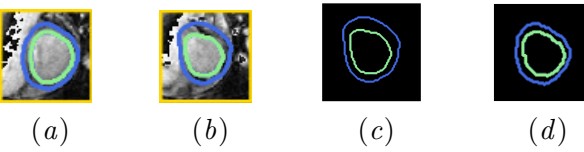

$(a)$ $(b)$ $(c)$ $(d)$

Figure 2: $(a)$ and $(b)$ zoomed moving and reference image, $(c)$ true segmented regions and $(d)$ warped segmented regions using estimated displacement field.

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
