# OpenReview forum: "Registration by tracking for sequential 2D imaging"
_MIDL.io/2021/Conference/Short — Submitted to MIDL 2021_

### Official Review · Reviewer_GT3M · 2021-04-20

**Confidence:** 4
**Final Rating:** 2

**Summary:**

This short paper tackles the problem of modality-agnostic image registration. The proposed solution is based on multiple instances of a visual tracking algorithm, and on a deep neural network with normalised convolutional layers. The first are used to compute the translations of pre-defined points between consecutive images, the second to refine the sparse displacements computed by the trackers into dense displacement vectors.

**Strengths:**

- **Motivation**. The problem of modality-agnostic image registration is relevant.
- **Idea**. The idea of employing DCF-based trackers to compute image translations is interesting. Also the employment of a neural network with normalised convolutional layers is appealing.
- **Presentation**. The paper is well written in general.
- **Code**. Code is made available.

**Weaknesses:**

- **Claim**. The paper claims the methodology is modality agnostic. However, there is no mention in the paper about which modalities the solution has been tested on. I guess on MRIs given the dataset linked in the "Data Set Url" field. This is not sufficient to claim that the solution generalises to other medical imaging modalities. Indeed, such a fact should be proven showing potential application to at least another modality.
- **Lack of important descriptions**. In my opinion, the paper focuses too much on explaining the working mechanism of DCFs, while not describing how the employed trackers have been exploited. In particular, the authors did not mention how the image points and respective regions to track have been determined. Furthermore, it is not clear in what consist the input of the neural network.
- **Validation**. The "Results" section presents just a qualitative example which is hard to understand since it has no meaningful description.  There is no mention about how the experimental validation was performed. Moreover, the authors could have reported some preliminary quantitative results achieved by the proposed solution.


**Deanonymize Review:**

no

**Detailed Comments:**

Please see the Weaknesses section.

**Justification Of The Rating:**

The problem of modality-agnostic image registration is important. The idea of employing DCFs for computing translations and of normalised convolutions is interesting. However, the description of the implemented solution lacks important details. Finally, the experimental results presented are not sufficient nor meaningful to really understand the potential of this idea.

**Paper Type:**

methodological development

**Special Issue:**

no

---

### Official Review · Reviewer_7NE2 · 2021-05-01

**Confidence:** 5
**Final Rating:** 2

**Summary:**

This paper addresses the problem of improving image registration for 2D sequences. In particular, authors use a sparse-to-dense scheme to mitigate the computational cost of a dense technique. The core of the technique is the use of discriminative correlation filter (DCF). There is no empirical support, experimental wise, of the technique.

**Strengths:**

--  The authors provide a good motivation for the problem at hand. Moreover, the use of a sparse-to-dense scheme for registration is indeed of interest for the community. The contribution of the paper is more the  application rather than technical which still is valid.



**Weaknesses:**

--  The authors fail to provide the big picture of the technicalities. Therefore, it is hard to appreciate the level of contribution

-- Likewise, authors provide very initial experiments that are not explained.  Authors fail to describe the main advantages of the technique wrt the body of literature.


**Deanonymize Review:**

no

**Detailed Comments:**

There are two major points that turn down the paper.


Firstly, authors fail to offer a good big picture of the technical part, which makes it hard to appreciate the level of contribution. Instead of Fig 1, it would be better to provide an explicit description and motivation of the technique. Authors should strongly argue and improve the claims on the level of novelty in the paper.

Secondly, authors attempted to provide an initial visualisation output of the technique. However, there are no discussion on the findings and how those visualisations fit into the big picture of the problem. Moreover, there are no discussion on the advantages in terms of computational time (which is one of the motivation when using a sparse-to-dense scheme) and performance wise.

Overall, the current version needs major updates to offer to the reader the big picture of the technical contribution and the findings from the provided experimental results. This version needs major updates.


**Justification Of The Rating:**

Overall, the current version needs major updates to offer to the reader the big picture of the technical contribution and the findings from the provided experimental results. This version needs major updates.


**Paper Type:**

validation/application paper

**Special Issue:**

no

---

### Meta-Review · Area_Chair_B3uD · 2021-05-07

**Recommendation:** Reject
**Confidence:** 5

**Metareview:**

The reviewers agree that there are important shortcomings, in particular in the experiments.

---

### Decision · Program_Chairs · 2021-05-11

Reject